# Gradient Inversion with Generative Image Prior

**Jinwoo Jeon**[1*]**, Jaechang Kim**[2*]**, Kangwook Lee**[3]**, Sewoong Oh**[4]**, Jungseul Ok**[1,2]

[1] Department of Computer Science & Engineering, Pohang University of Science and Technology
[2] Graduate School of Artificial Intelligence, Pohang University of Science and Technology
[3] Department of Electrical and Computer Engineering, University of Wisconsin-Madison, Madison
[4] Paul G. Allen School of Computer Science & Engineering, University of Washington

## Abstract

Federated Learning (FL) is a distributed learning framework, in which the local data never leaves clients' devices to preserve privacy, and the server trains models on the data via accessing only the gradients of those local data. Without further privacy mechanisms such as differential privacy, this leaves the system vulnerable against an attacker who inverts those gradients to reveal clients' sensitive data. However, a gradient is often insufficient to reconstruct the user data without any prior knowledge. By exploiting a generative model pretrained on the data distribution, we demonstrate that data privacy can be easily breached. Further, when such prior knowledge is unavailable, we investigate the possibility of learning the prior from a sequence of gradients seen in the process of FL training. We experimentally show that the prior in a form of generative model is learnable from iterative interactions in FL. Our findings strongly suggest that additional mechanisms are necessary to prevent privacy leakage in FL.

## 1 Introduction

Federated learning (FL) is an emerging framework for distributed learning, where central server aggregates model updates, rather than user data, from end users [5, 17]. The main premise of federated learning is that this particular way of distributed learning can protect users' data privacy as there is no explicit data shared by the end users with the central server.

However, a recent line of work [34, 31, 9, 29] demonstrates that one may recover the private user data used for training by observing the gradients. This process of recovering the training data from gradients, so-called *gradient inversion*, poses a huge threat to the federated learning community, as it may imply the fundamental flaw of its main premise.

Even more worryingly, recent works suggest that such gradient inversion attacks can be made even stronger if certain side-information is available. For instance, Geiping et al. [9] show that if the attacker knows a prior that user data consists of natural images, then the gradient inversion attack can leverage such prior, achieving a more accurate recovery of the user data. Another instance is when batch norm statistics are available at the attacker in addition to gradients. This can actually happen if the end users share their local batch norm statistics as in [17]. Yin et al. [29] show that such batch normalization statistics can significantly improve the strength of the gradient inversion attack, enabling precise recovery of high-resolution images.

In this paper, we systematically study how one can maximally utilize and even obtain the prior information when inverting gradients. We first consider the case that the attacker has a generative model pretrained on the exact or approximate distribution of the user data as a prior. For this, we propose an efficient gradient inversion algorithm that utilizes the generative model prior. In

---

* equal contribution

35th Conference on Neural Information Processing Systems (NeurIPS 2021).

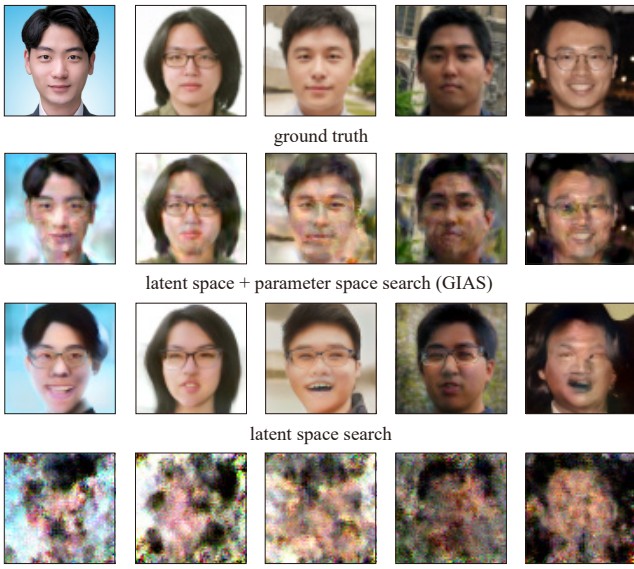

ground truth

latent space + parameter space search (GIAS)

latent space search

existing method [9]

Figure 1: *An example showing the superiority of GIAS compared to existing method.* Images of the authors are reconstructed from gradients by exploiting a generative model pretrained on human face images.

particular, the algorithm consists of two steps, in which the first step searches the latent space (of lower dimension) defined by the generative model instead of the ambient input space (of higher dimension), and then the second step adapts the generative model to each input given the gradient. Each step provides substantial improvement in the reconstruction. We name the algorithm as gradient inversion in alternative spaces (GIAS). Figure 1 represents reconstruction results with the proposed method and existing one.

We then consider a realistic scenario in which the user data distribution is not known in advance, and thus the attacker needs to learn it from gradients. For this scenario, we develop a meta-learning framework, called gradient inversion to meta-learn (GIML), which learns a generative model on user data from observing and inverting multiple gradients computed on the data, e.g. across different FL epochs or participating nodes. Our experimental results demonstrate that one can learn a generative model via GIML and reconstruct data by making use of the learned generative model.

This implies a great threat on privacy leakage in FL since our methods can be applied for any data type in most FL scenarios unless a specialized architecture prevents the gradient leakage explicitly, e.g., [18].

Our main contributions are as follows:

- We introduce GIAS that fully utilizes a pretrained generative model to invert gradient. In addition, we propose GIML which can train generative model from gradients only in FL.

- We demonstrate significant privacy leakage occurring by GIAS with a pretrained generative model in various FL scenarios which are challenging to other existing methods, e.g., [9, 29].

- We experimentally show that GIML can learn a generative model on the user data from only gradients, which provides the same level of data recovery with a given pretrained model. To our best knowledge, GIML is the first capable of learning explicit prior on a set of gradient inversion tasks.

- We note that a gradient inversion technique defines a standard on defence mechanism in FL for privacy [28]. By substantiating that our proposed methods are able to break down defense mechanisms that were safe according to the previous standard, we give a strong warning to the FL community to use a higher standard defined by our attack methods, and raise the necessity of a more conservative choice of defense mechanisms.

## 2 Related work

**Privacy attacks in FL.**   Early works [19, 24] investigate membership inference from gradients to check the possibility of privacy leakage in FL. Phong et al. [21] demonstrate that it is possible to reconstruct detailed input image when FL trains a shallow network such as single-layer perceptron. Fan et al. [7] and Zhu and Blaschko [32] consider a wider class of learning model and propose an analytical approach solving a sequence of linear systems to reveal the output of each layer recursively. To study the limit of the gradient inversion in practical scenarios of training deep networks via FL, a sequence of effort has been made formulating optimization problem to minimize discrepancy comparing gradients from true data and reconstructed data [9, 27, 29, 31, 34].

**Gradient inversion with prior.**   The optimization-based approaches are particularly useful as one can easily utilize prior knowledge by adding regularization terms, e.g., total variation [27, 9] and BN statistics [29], or changing discrepancy measure [9] . In [29], a privacy attack technique using a generative model is introduced. They however require a pretrained model, while we propose a meta learning framework training generative model from gradients only. In addition, our method of inverting gradient maximally exploit a given generative model by alternating search spaces, which are analogous to the state-of-the-art GAN inversion techniques [3, 4, 33].

**Generative model revealing private data.**   Training a generative model with transmitted gradients also demonstrates privacy leakage in FL. Hitaj et al. [11] introduce an algorithm to train a GAN regarding shared model in FL framework as a discriminator. Wang et al. [27] use reconstructed data from gradient to train a GAN. Those works require some auxiliary dataset given in advance to enable the training of GAN, while we train a generative model using transmitted gradients only. Also, we not only train a generative model but also utilize it for reconstruction, while the generative models in [11, 27] are not used for the reconstruction. Hence, in our approach, the generative model and reconstruction can be improved interactively to each other as shown in Figure 6. In addition, [27] is less sample-efficient than ours in a sense that they use gradients to reconstruct images and then train a generative model with the reconstructed images, i.e., if the reconstruction fails, then the corresponding update of the generative model fails too, whereas we train the generative model directly from gradients.

## 3 Problem formulation

In this section, we formally describe the gradient inversion (GI) problem. Consider a standard supervised learning for classification, which optimizes neural network model $f_\theta$ parameterized by $\theta$ as follows:

$$\min_\theta \sum_{(x,y)\in\mathcal{D}} \ell(f_\theta(x), y) , \tag{1}$$

where $\ell$ is a point-wise loss function and $\mathcal{D}$ is a dataset of input $x \in \mathbb{R}^m$ and label $y \in \{0,1\}^L$ (one-hot vector). In federated learning framework, each node reports the gradient of $\ell(f_\theta(x), y)$ for sampled data $(x, y)$'s instead of directly transferring the data. The problem of inverting gradient is to reconstruct the sampled data used to compute the reported gradient. Specifically, when a node computes the gradient $g$ using a batch $\{(x_1^*, y_1^*), ..., (x_B^*, y_B^*)\}$, i.e., $g = \frac{1}{B}\sum_{j=1}^B \nabla\ell(f_\theta(x_j^*), y_j^*)$, we consider the following problem of inverting gradient:

$$\min_{\substack{(x_1,y_1),\cdots,(x_B,y_B) \\ \in\mathbb{R}^m\times\{0,1\}^L}} d\left(\frac{1}{B}\sum_{j=1}^B \nabla\ell(f_\theta(x_j), y_j), g\right) , \tag{2}$$

where $d(\cdot, \cdot)$ is a measure of the discrepancy between two gradient, e.g., $\ell_2$-distance [34, 29] or negative cosine similarity [9]. It is known that label $y$ can be almost accurately recovered by simple methods just observing the gradient at the last layer [31, 29], while reconstructing input $x$ remains still challenging as it is often under-determined even when the true label is given. For simplicity, we hence focus on the following minimization to reveal the inputs from the gradient given the true labels:

$$\min_{x_1,...,x_B\in\mathbb{R}^m} c\left(x_1, ..., x_B; \theta, g\right) , \tag{3}$$

where we denote by $c\left(x_1, ..., x_B; \theta, g\right)$ the cost function in (2) given $y_j = y_j^*$ for each $j = 1, ..., B$.

## 4 Methods

The key challenge of inverting gradient is that solving (2) is often under-determined, i.e., a gradient contains only insufficient information to recover data. Such an issue is observed even when the dimension of gradient is much larger than that of input data. Indeed, Zhu and Blaschko [32] show that there exist a pair of different data having the same gradient, so called twin data, even when the learning model is large. To alleviate this issue, a set of prior knowledge on the nature of data can be considered.

When inverting images, Geiping et al. [9] propose to add the total variation regularization $R_{\text{TV}}(x)$ to the cost function in (3) since neighboring pixels of natural images are likely to have similar values. More formally,

$$R_{\text{TV}}(x) := \sum_{(i,j)} \sum_{(i',j') \in \partial(i,j)} \|x(i,j) - x(i',j')\|^2 , \tag{4}$$

where $\partial(i,j)$ is the set of neighbors of $(i,j)$. This method is limited to the natural image data.

For general type of data, one can consider exploiting the batch normalization (BN) statistics from nodes. This is available in the case that the server wants to utilize batch normalization (BN) in FL, and thus collects the BN statistics (mean and variance) of batch from each node, in addition, with every gradient report [17]. To be specific, Yin et al. [29] propose to employ the regularizer $R_{\text{BN}}(x_1, ..., x_B; \theta)$ which quantifies the discrepancy between the BN statistics of estimated $x_j$'s and those of true $x_j^*$'s on each layer of the learning model. More formally,

$$R_{\text{BN}}(x_1, ..., x_B; \theta) := \sum_l \|\mu_l - \mu_{l,\text{exact}}\|_2 + \|\sigma_l^2 - \sigma_{l,\text{exact}}^2\|_2,$$

where $\mu_l(x_1, ..., x_B; \theta)$ and $\sigma_l^2(x_1, ..., x_B; \theta)$ (resp. $\mu_{l,\text{exact}}(x_1^*, ..., x_B^*; \theta)$ and $\sigma_{l,\text{exact}}^2(x_1^*, ..., x_B^*; \theta)$) are the mean and variance of $l$-th layer feature maps for the estimated batch $x_1, ..., x_B$ (resp. the true batch $x_1^*, ..., x_B^*$) given $\theta$. This is available only if clients agree to report their exact BN statistics at every round. But not every FL framework report BN statistics [15, 2]. In that case, Yin et al. [29] also propose to use the BN statistics over the entire data distribution as a proxy of the true BN statistics, and reports that the gain from the approximated BN statistics is comparable to that from the exact ones. The applicability of $R_{\text{BN}}$ with the approximated BN statistics is still limited as the proxy needs to be additionally recomputed over the entire data distribution at every change of $\theta$. However, this demonstrates the significant impact of knowing the data distribution in the gradient inversion and motivates our methods using and learning a generative model on the user data, described in what follows.

### 4.1 Gradient inversion with trained generative model

Consider a decent generative model $G_w : \mathbb{R}^k \mapsto \mathbb{R}^m$ trained on the approximate (possibly exact) distribution of user data $\mathcal{D}$ such that $x^* \approx G_w(z^*)$ for $(x^*, \cdot) \in \mathcal{D}$ and its latent code $z^* = \arg\min_z \|G_w(z) - x^*\|$. To fully utilize such a pretrained generative model, we propose gradient inversion in alternative spaces (GIAS), of which pseudocode is presented in Appendix A, which performs latent space search over $z$ and then parameter space search over $w$. We also illustrate the overall procedure of GIAS in Figure 2.

**Latent space search.** Note that the latent space is typically much smaller than the ambient input space, i.e., $k \ll m$, for instances, DCGAN [25] of $k = 100$ and StyleGAN [12] of $k = 512 \times 16$ for image data of $m = (\text{width}) \times (\text{height}) \times (\text{color})$ such as $32 \times 32 \times 3$, $256 \times 256 \times 3$, or larger. Using such a pretrained generative model with $k \ll m$, the under-determined issues of (3) can be directly mitigated by narrowing down the searching space from $\mathbb{R}^m$ to $\{G_w(z) : z \in \mathbb{R}^k\}$. Hence, GIAS first performs the latent space search in the followings:

$$\min_{z_1, ..., z_B \in \mathbb{R}^k} c\left(G_w(z_1), ..., G_w(z_B)\right) . \tag{5}$$

Considering a canonical class of neural network model $f_\theta$, we can show that the reconstruction of $x^*$ by latent space search in (5) aligns with that by input space search in (3) if the generative model $G_w$ approximates input data with small enough error.

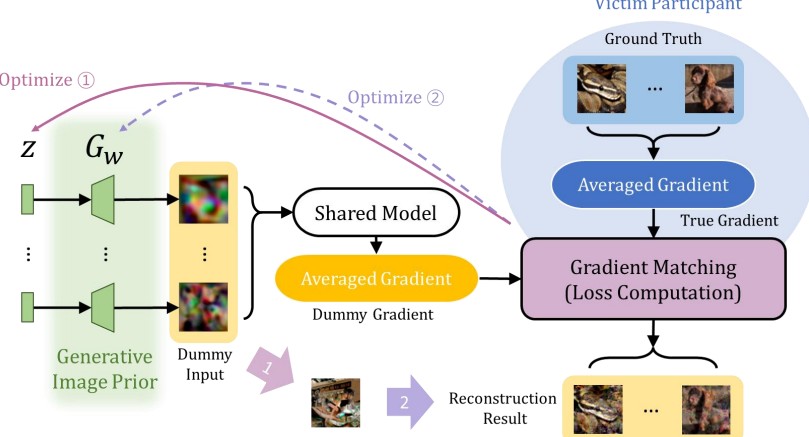

Figure 2: *An overview of GIAS*. GIAS optimizes a latent code $z$ and generative model parameters $w$ to reconstruct the data which matches the gradient.

**Property 1.** *For an input data $x^* \in [0,1]^m$ consider the gradient inversion problem of minimizing cost $c$ in* (3)*, where a canonical form of deep learning for classification is considered and the discrepancy measure $d$ is $\ell_2$-distance. Suppose that it has the unique global minimizer at $x^*$. Let $\varepsilon \geq 0$ be the approximation error bound on $x^*$ for generative model $G_w : [0,1]^k \mapsto [0,1]^m$ Then, there exists $\delta(\varepsilon) \geq 0$ such that for any $z^* \in \arg\min_z c(G_w(z))$,*

$$\|G_w(z^*) - x^*\| \leq \delta(\varepsilon) ,\tag{6}$$

*of which upper bound $\delta(\varepsilon) \to 0$ as $\varepsilon \to 0$.*

A rigorous statement of Property 1 and its proof are provided in Appendix B, where we prove and use that the cost function is continuous around $x^*$ under the assumptions. This property justifies solving the latent space search in (5) for FL scenarios training neural network model while it requires an accurate generative model.

**Parameter space search.** Using the latent space search only, there can be inevitable reconstruction error due to the imperfection of generative model. This is mainly because we cannot perfectly prepare the generative model for every plausible data in advance. Similar difficulty of the latent space search has been reported even when inverting GAN [33, 3, 4] for plausible but new data directly, i.e., $\min_z \|G_w(z) - x^*\|$ given $x^*$, rather than inverting gradient. Bau et al. [3] propose an instance-specific model adaptation, which slightly adjusts the model parameter $w$ to (a part of source image) $x^*$ after obtaining a latent code $z^*$ for $x^*$. Inspired by such an instance-specific adaptation, GIAS performs the following parameter space search over $w$ preceded by the latent space search over $z$:

$$\min_{w_1,...,w_B} c\left(G_{w_1}(z_1), ..., G_{w_B}(z_B)\right) ,\tag{7}$$

where $z_1, \ldots, z_B$ are obtained from (5).

**Remark.** We propose the optimization over $w$ followed by that over $z$ sequentially This is to maximally utilize the benefit of mitigating the under-determined issue from reducing the searching space on the pretrained model. However, the benefit would be degenerated if $z$ and $w$ are optimized jointly or $w$ is optimized first. We provide an empirical justification on the proposed searching strategy in Section 5.1.

We perform each search in GIAS using a standard gradient method to the cost function directly. It is worth noting that those optimizations (5) and (7) with generative model can be tackled in a recursive manner as R-GAP [32] reconstructs each layer from output to input. We provide details and performance of the recursive procedure in Appendix C, where employing generative model improves the inversion accuracy of R-GAP substantially, while R-GAP apparently suffers from an error accumulation issue when $f_\theta$ is a deep neural network.

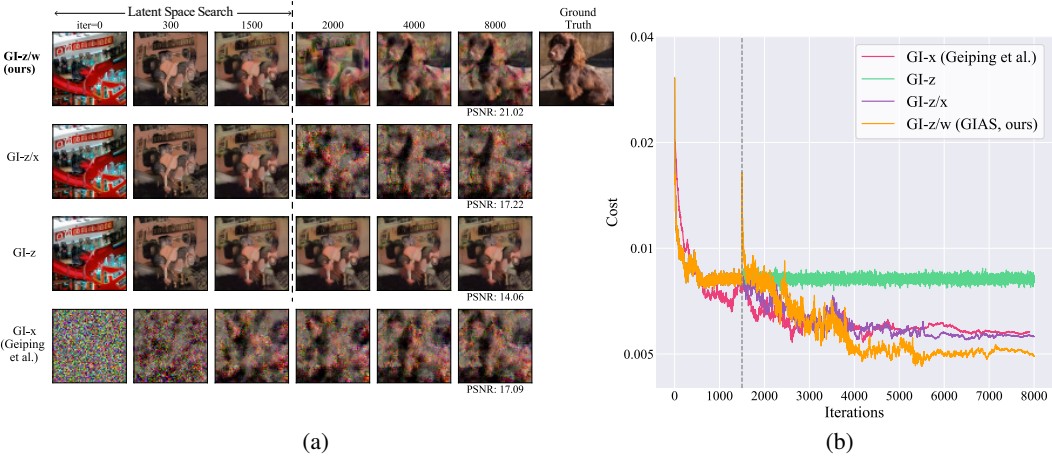

Figure 3: *Comparison of different searching spaces.* (a) Each row shows reconstructed images of different optimization domains. The first three rows share the same latent space search of $1,500$ iterations, and GI-$z/w$ is verified to be the best option to fully exploits the knowledge inside the generative model. (b) Cost function over iterations of different optimization domains.

## 4.2 Gradient inversion to meta-learn generative model

For the case that pretrained generative model is unavailable, we devise an algorithm to train a generative model $G_w$ for a set $\mathcal{S} = \{(\theta_i, g_i)\}$ of gradient inversion tasks. Since each inversion task can be considered as a small learning task to adapt generative model per data, we hence call it gradient inversions to meta-learn (GIML). The detailed procedure of GIML is presented in Appendix A. We start with an arbitrary initialization of $w$, and iteratively update toward $w'$ from a variant of GIAS for $N$ tasks sub-sampled from $\mathcal{S}$, which is different than multiple applications of GIAS for each task in two folds: (i) $\ell_2$-regularization in latent space search; and (ii) an integrated optimization on model parameter. The variant first finds optimal latent codes $\boldsymbol{z}_i^* = (z_{i1}^*, ..., z_{iB}^*)$ for each task $i$ with respect to the same cost function of GIAS but additional $\ell_2$-regularization. Note that the latent space search with untrained generative model easily diverges. The $\ell_2$-regularization is added to prevent the divergence of $\boldsymbol{z}_i^*$. Once we obtained $\boldsymbol{z}_i^*$'s, $w'$ is computed by few steps of gradient descents for an integrated parameter search to minimize $\sum_i c(G_{w'}(z_{i1}^*), ..., G_{w'}(z_{iB}^*); \theta_i, g_i)$. This is because in GIML, we want meta information $w$ to help GIAS for each task rather than solving individual tasks, while after performing GIML to train $w$, we perform GIAS to invert gradient with the trained $w$. This is analogous to the Reptile in [20].

## 5 Experiments

**Setup.** Unless stated otherwise, we consider the image classification task on the validation set of ImageNet [22] dataset scaled down to $64 \times 64$ pixels (for computational tractability) and use a randomly initialized ResNet18 [10] for training. For deep generative models in GIAS, we use StyleGAN2 [13] trained on ImageNet. We use a batch size of $B = 4$ as default and use the negative cosine to measure the gradient dissimilarity $d(\cdot, \cdot)$. We present detailed setup in Appendix H. Our experiment code is available at `https://github.com/ml-postech/gradient-inversion-generative-image-prior`.

**Algorithms.** We evaluate several algorithms for the gradient inversion (GI) task in (3). They differ mainly in which spaces each algorithm searches over: the input $x$, the latent code $z$, and/or the model parameter $w$. Each algorithm is denoted by GI-$(\cdot)$, where the suffix indicates the search space(s). For instances, GI-$z/w$ is identical to the proposed method, GIAS, and GI-$x$ is the one proposed by Geiping et al. [9].

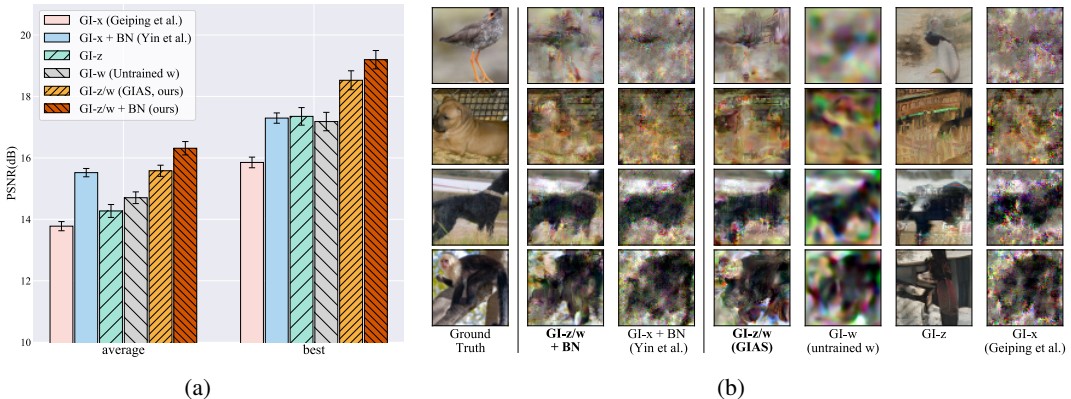

(a)            (b)

Figure 4: *Comparison of state-of-the-art models and ours*. Replacing GI-$x$ with GI-$z/w$ (GIAS) regardless of using BN [29] or not [9] provides substantial improvement in the reconstruction accuracy. (a) Average PSNR and best PSNR in a batch throughout the experiments. (b) An ablation study and comparison of reconstruction results with our models and state-of-the-art models. We highlight the proposed models in **bold**.

Table 1: Comparison of our methods with state-of-the-art methods. Adding our method makes performance improvement versus two baseline methods. PSNR, SSIM, and LPIPS[30] are used to evaluate reconstruction results. We highlight the best performances in **bold**.

| Method | GI-$x$ [9] | GI-$z$ (ours) | GI-$w$ (ours) | GI-$z/w$ (GIAS, ours) | GI-$x$+BN [29] | GI-$z/w$+BN (ours) |
|---|---|---|---|---|---|---|
| PSNR ↑ | 13.78 | 14.27 | 14.70 | **15.58** | 15.52 | **16.31** |
| SSIM ↑ | 0.2542 | 0.3106 | 0.3519 | **0.3895** | 0.3513 | **0.4311** |
| LPIPS ↓ | 0.4376 | 0.3233 | 0.5121 | **0.3023** | 0.3645 | **0.2861** |

## 5.1 Justification of GIAS design

We first provide an empirical justification of the specific order of searching spaces in GIAS (corresponding to GI-$z/w$) to fully utilize a pretrained generative model. To do so, we provide Figure 4b comparing algorithms with different searching spaces: GI-$z/w$, GI-$z/x$, GI-$z$, and GI-$x$, of which the first three share the same latent space search over $z$ for the first $1,500$ iterations. As shown in Figure 3(a), the latent space search over $z$ quickly finds plausible image in a much shorter number of iterations than GI-$x$, while it does not improve after a certain point due to the imperfection of pretrained generative model. Such a limitation of GI-$z$ is also captured in Figure 3(b), where the cost function of GI-$z$ is not decreasing after a certain number of optimization steps. To further minimize the cost function, one alternative to GI-$z/w$ (GIAS) is GI-$z/x$, which can further reduce the loss function whereas the parameter search in GI-$z/w$ seems to provide more natural reconstruction of the image than GI-$z/x$. The superiority of GI-$z/w$ over GI-$z/x$ may come from that the parameter space search exploits an implicit bias from optimizing a good architecture for expressing images, c.f., deep image prior [26]. In Appendix E and Figure 1, we also present the same comparison on FFHQ (human-face images) [12] where diversity is much smaller than that of ImageNet. On such a less diverse dataset, the distribution can be easily learned, and the gain from training a generative model is larger.

## 5.2 The gain from fully exploiting pretrained generative model

**Comparison with state-of-the-art models.** Our method can be easily added to previous methods [9, 29]. In Table 1 and Figure 4, we compare the state-of-the-art methods both with and without the proposed generative modelling. In Table 1, comparing GI-$x$ to GI-$z/w$ and GI-$x$ + BN to GI-$z/w$ + BN, adding the proposed generative modelling provides additional gain in terms of all the measures (PSNR, SSIM, LPIPS) of reconstruction quality. GI-$z/w$ without BN has lower reconstruction error than GI-$x$ + BN, which is the method of [29]. This implies that the gain from the generative model is comparable to that from BN statistics. However, while the generative model only requires a global (and hence coarse) knowledge on the *entire dataset*, BN statistics are local to the batch in hand and hence requires significantly more detailed information on the *exact batch* used to compute gradient.

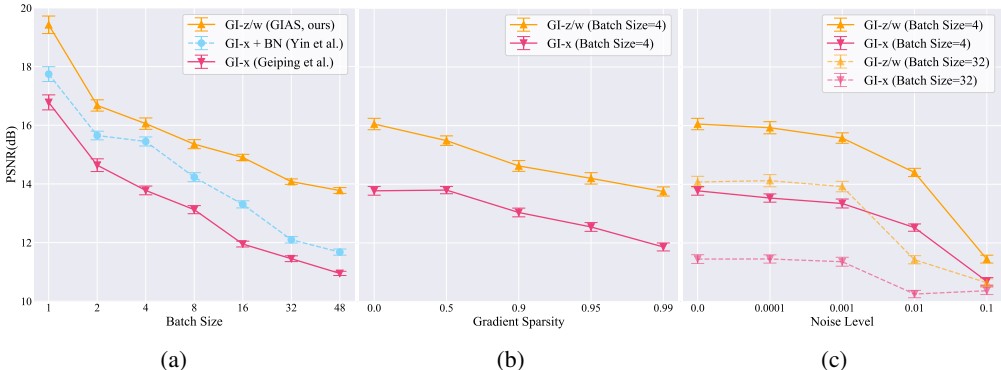

Figure 5: *Comparison of state-of-the-art models and GI-$z/w$ with varying difficulties.* Larger batch size, higher sparsity, and larger gradient noise increases reconstruction difficulty. GI-$z/w$ always surpasses GI-$x$ thanks to the pretrained generative model. All subfigures share the y-axis.

As shown in Figure 4, the superiority of our method compared to the others is clear in terms of the best-in-batch performance than the average one, where the former is more suitable to show actual privacy threat in the worst case than the latter. It is also interesting to note that GI-$w$ with untrained $w$ provides substantial gain compared to GI-$x$. This may imply that there is a gain of the implicit bias, c.f., [26], from training the architecture of deep generative model.

**Evaluation against possible defense methods**   We evaluate the gain of using a generative model for various FL scenarios with varying levels of difficulty in the inversion. As batch size, gradient sparsity[1] [28] and gradient noise level increase, the risk of having under-determined inversion increases and the inversion task becomes more challenging. Figure 5 shows that for all the levels of difficulty, the generative model provides significant gain in reconstruction quality. In particular, the averaged PSNR of GI-$x$ with a batch size of 4 is comparable to that of GI-$z/w$ with a batch size 32. It is also comparable to that of GI-$z/w$ with a gradient sparsity of 99%. To measure the impact of the noisy gradient, we experimented gradient inversion with varying gaussian noise level in aforementioned settings. Figure 5(c) shows that adding enough noise to the gradient can mitigate the privacy leakage. GI-$z/w$ with a noise level of $0.01$, which is relatively large, still surpasses GI-$x$ without noise. A large noise of $0.1$ can diminish the gain of exploiting a pretrained generative model. However, the fact that adding large noise to the gradient slows down training makes it difficult for FL practitioners to choose suitable hyperparameters. The results imply our method is more robust to defense methods against gradient inversion, but can be blocked by a high threshold. Note that our results of gradient sparsity and gradient noise implies the Differential Privacy(DP) is still a valid defense method, when applied with a more conservative threshold. For more discussion about possible defense methods in FL framework, see Appendix F.

## 5.3   Learning generative model from gradients

We demonstrate the possibility of *training* a generative model only with gradients. For computational tractability, we use DCGAN and images from FFHQ [12] resized to 32x32. We generate a set of gradients from 4 rounds of gradient reports from 200 nodes, in which node computes gradient for a classification task based on the annotation provided in [6]. From the set of gradients, we perform GIML to train a DCGAN to potentially generate FFHQ data.

Figure 6 shows the evolution of generative model improves the reconstruction quality when performing either GI-$z$ and GI-$z/w$. We can clearly see the necessity of parameter space search. Figure 6(a) shows that the quality of images from the generative model is evolving in the training process of GIML. As the step $t$ of GIML increases, the generative model $G_{w^{(t)}}(z)$ for arbitrary $z$ outputs more plausible image of human face. When using generative model trained on wrong dataset (CIFAR10), GI-$z$ completely fails at recovering data.

---

[1]Having gradient sparsity 0.99% implies that we reconstruct data from 1% of the gradient after removing 99% elements with the smallest magnitudes at each layer.

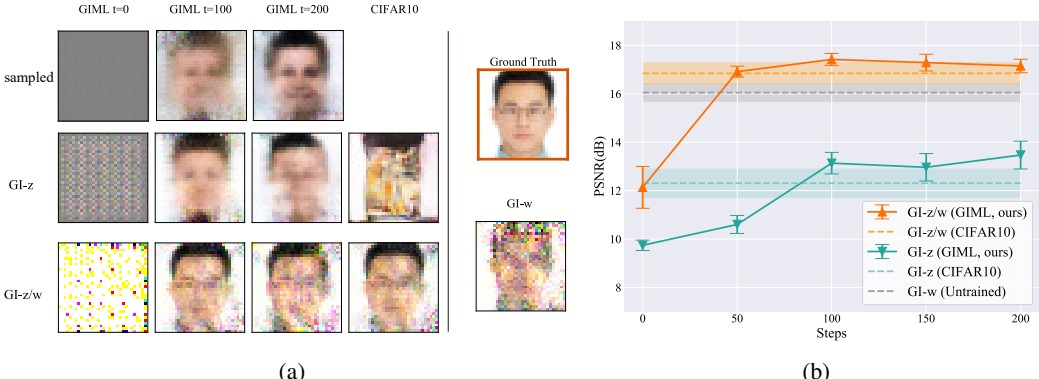

(a)  (b)

Figure 6: *Qualitative and quantitative result of GIML.* (a) Results validating generative model trained with GIML. Images on the first row are sampled from different GIML training steps. The same latent code $z$ was used to sample images in same rows. Images on the second row and third row are results of GI-$z$ and GI-$z/w$ using generative model trained with GIML and pretrained model which is trained with CIFAR10 images. Experiments were done with gradient sparsity 0.95 for comparison in difficult setting. Last column represents the ground truth image and result of GI-$w$ with untrained model. (b) A comparison of GIAS with meta-learned generative model and GIAS using improper generative model. Proper generative model boosts GIAS performance.

In Figure 6(b), as GIML iteration step increases, the performance of GI-$z$ and GI-$z/w$ with GIML surpass GI-$z$ and GI-$z/w$ with wrong prior knowledge. GI-$z/w$ using generative model trained on wrong dataset and GI-$w$ which starts with an untrained generative model show lower averaged PSNR compared to GI-$z/w$ with GIML. GI-$z/w$ with GIML to train generative model on right data shows the best performance in terms of not only quality (Figure 6) but also convergence speed. We provide a comparison of the convergence speed in Appendix G.

## 6  Conclusion

We propose GIAS fully exploit the prior information on user data from a pretrained generative model when inverting gradient. We demonstrate significant privacy leakage using GIAS with pretrained generative model in various challenging scenarios, where our method provides substantial gain additionally to any other existing methods [9, 29]. In addition, we propose GIML which can train a generative model using only the gradients seen in the FL classifier training. We experimentally show that GIML can meta-learn a generative model on the user data from only gradients, which improves the quality of each individual recovered image. To our best knowledge, GIML is the first capable of learning explicit prior on a set of gradient inversion tasks.

## Acknowledgments

This work was partly supported by Institute of Information & communications Technology Planning & Evaluation (IITP) grant funded by the Korea government (MSIT) (No. 2019-0-01906, Artificial Intelligence Graduate School Program (POSTECH)) and (No. 2021-0-00739, Development of Distributed/Cooperative AI based 5G+ Network Data Analytics Functions and Control Technology). Jinwoo Jeon and Jaechang Kim were supported by the Institute of Information & Communications Technology Planning & Evaluation (IITP) grant funded by Korea(MSIT) (2020-0-01594, PSAI industry-academic joint research and education program). Kangwook Lee was supported by NSF/Intel Partnership on Machine Learning for Wireless Networking Program under Grant No. CNS-2003129 and NSF Award DMS-2023239. Sewoong Oh acknowledges funding from NSF IIS-1929955, NSF CCF 2019844, and Google faculty research award.

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
