# Appendix

## A    Detailed algorithms

---
**Algorithm 1** Gradient Inversion in Alternative Spaces (GIAS)

---
**Require:** learning model $f_\theta$; target gradient $g = \nabla_\theta \sum_{j=1}^B \ell(f_\theta(x_j^*), y_j^*)$ to be inverted; batch size B; pre-trained generative model $G_w$;

1: Initialize $\boldsymbol{z} := (z_1, ..., z_B)$ randomly
2: Find $\boldsymbol{z} \leftarrow \arg\min_{\boldsymbol{z}} c(G_w(z_1), ..., G_w(z_B))$
                                                     // Latent space search
3: Set $\boldsymbol{w} := (w_1, \ldots, w_B) \leftarrow (w, \ldots, w)$
4: Find $\boldsymbol{w} \leftarrow \arg\min_{\boldsymbol{w}} c(G_{w_1}(z_1), \ldots, G_{w_B}(z_B))$
                                                     // Parameter space search
5: Return result: $G_{w_1'}(z_1), \ldots, G_{w_B'}(z_B)$

---

---
**Algorithm 2** Gradient Inversion to Meta-Learn generative model (GIML)

---
**Require:** inversion task set $\mathcal{S}$; task batch size $N$; data batch size $B$ (per gradient); number of local iterations $\tau$; $z$-regularizer coefficient $\lambda$; step sizes $\alpha, \beta$;
1: Initialize $w$ randomly
2: **while** not done **do**
3:    Sample a batch of inversion tasks $(\theta_1, g_1), ..., (\theta_N, g_N)$ from $\mathcal{S}$
4:    $w' \leftarrow w$
5:    **for all** $i = 1, \ldots, N$ **do**
6:        $\boldsymbol{z}_i^* \leftarrow \mathrm{argmin}_{\boldsymbol{z}_i} c(G_{w'}(z_{i1}), \ldots, G_{w'}(z_{iB}); \theta_i, g_i) + \lambda \sum_j \|z_{ij}\|_2$    // Regularized latent space search
7:    **end for**
8:    **for all** $t = 1, \ldots, \tau$ **do**
9:        $w' \leftarrow w' - \alpha \nabla_{w'} \sum_i c\left(G_{w'}(z_{i1}^*), \ldots, G_{w'}(z_{iB}^*); \theta_i, g_i\right)$   // Meta parameter space search
10:   **end for**
11:   Update $w \leftarrow w - \beta(w - w') = (1 - \beta)w + \beta w'$
12: **end while**

---

## B    Proof of Property 1

To prove Property 1, we first conclude the same statement of Property 1 assuming the inversion problem is continuous at $x^*$ (Lemma 1), and then show that the standard scenario described in Property 1 guarantees the desired continuity (Lemma 2). The canonical form of learning model mentioned in 1 is described by the assumptions of Lemma 2.

**Lemma 1** (An extension of Property 1 in the main text). *For an input data $x^* \in [0, 1]^m$, consider the gradient inversion problem of minimizing cost $c(x)$ in (3) where $c(x)$ is continuous. Suppose that it has the unique global minimizer at $x^*$. Let $\varepsilon \geq 0$ be the approximation error bound on $x^*$ for generative model $G_w : [0, 1]^k \mapsto [0, 1]^m$ with $k \leq m$, i.e., $\min_{z \in [0,1]^k} \|x^* - G_w(z)\| \leq \varepsilon$. Then, there exists $\delta(\varepsilon) \geq 0$ such that for any $z^* \in \arg\min_z c(G_w(z))$,*

$$\|G_w(z^*) - x^*\| \leq \delta(\varepsilon) , \tag{8}$$

*of which upper bound $\delta(\varepsilon) \to 0$ as $\varepsilon \to 0$.*

*Proof of Lemma 1.* From the assumptions that $x^*$ is the unique minimizer and $c(x)$ is continuous on $[0, 1]^m$, it follows that for $x \in [0, 1]^m$, if $c(x) \to c(x^*)$, then $x \to x^*$. This can be proved by contradiction. Then we have that for $\varepsilon > 0$, there exists $\delta(\varepsilon) > 0$ such that if $c(x) \leq \varepsilon$, then $\|x - x^*\| \leq \delta(\varepsilon)$ where $\delta(\varepsilon) \to 0$ as $\varepsilon \to 0$. From the continuity of $c(x)$, it is straightforward to check that $c(G_w(z^*(\varepsilon))) \to c(x^*)$ as $\varepsilon \to 0$. This completes the proof. $\quad\square$

**Lemma 2.** *For an input data $x^* \in \mathbb{R}^m$, consider the gradient inversion problem of minimizing cost $c$ in (3), where the learning model $f_\theta$ is a standard form of R-layer neural network $f_\theta(x) = \Theta_R \sigma_{R-1}(\Theta_{R-1} \sigma_{R-2}(...\Theta_1 x))$ with $\Theta_r \in \mathbb{R}^{m_r \times m_{r-1}}$ for each $r$, where $m_R = L$ and $m_0 = m$, and C1 (continuously differentiable) activation $\sigma$'s (e.g., sigmoid and exponential linear), loss function $\ell$ is C1 (e.g., logistic and exponential), and the discrepancy measure $d$ is $\ell_2$-distance. Then, the corresponding cost function $c(x)$ is continuous with respect to $x \in \mathbb{R}^m$.*

*Proof of Lemma 2.* Note that the standard model $f_\theta$ includes multi-layer perceptron or convolutional neural network. It is $C1$ since the composition of C$k^2$ functions is C$k$. Hence, the gradient is continuous w.r.t. $x$. In addition, the cost function $c(x)$ is continuous since the gradient and the choice of discrepancy measure are continuous. This concludes the proof. $\qquad\square$

The proof of Property 1 is straightforward from Lemmas 1 and 2.

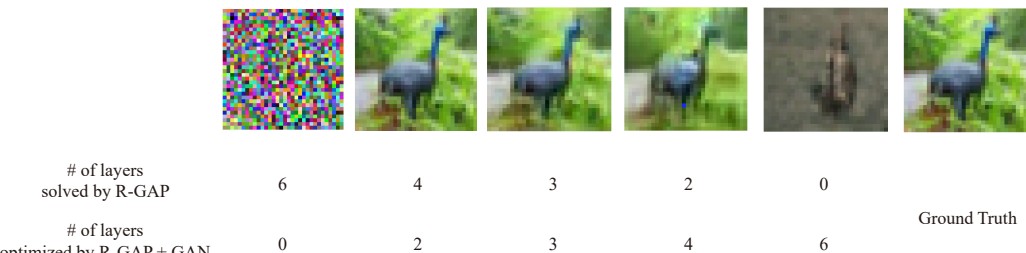

| # of layers solved by R-GAP | 6 | 4 | 3 | 2 | 0 | |
| # of layers optimized by R-GAP + GAN | 0 | 2 | 3 | 4 | 6 | Ground Truth |

Figure A1: Comparison of R-GAP and R-GAP with a generative model. The second convolution layer is rank-deficient and R-GAP should solve under-determined problem. An under-determined problem is solved by using generative model. However, the error per layer increases much faster than R-GAP.

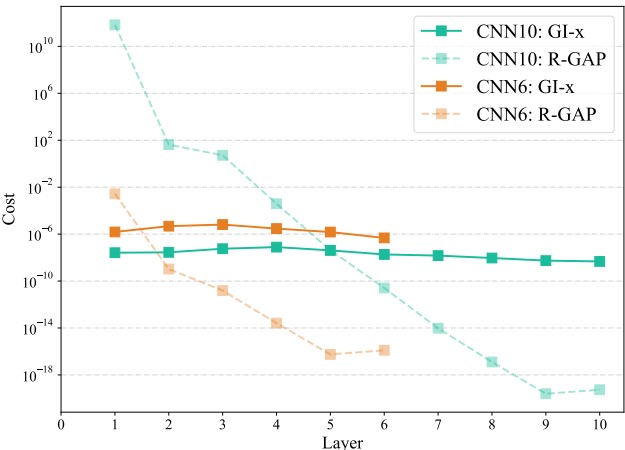

Figure A2: Layer-wise errors of convolution layers in gradient inversion attack results. layer-by-layer method shows cumulative exploding error. CNN6 denotes a neural network consists of six convolution layers and one FC layer. CNN10 denotes a neural network consists of ten convolution layers and one FC layer.

## C  Another method for gradient inversion: R-GAP [32]

In the main text, to solve the inversion problem in (2), we use gradient descent method directly to the cost function, while we alternate the searching space. Meanwhile, Zhu and Blaschko [32] propose another approach, called R-GAP (recursive gradient attack on privacy), to solve the optimization in

---
[2]the $k$-th derivative is continuous

(2), although it is limited to the case when the learning model $f_\theta$ is given as a standard form described in Lemma 2. R-GAP decomposes the optimization (2) into a sequence of linear programming to reconstruct the output of each layer except the last layer's, and then it solves them recursively from the penultimate layer to the input layer. The linear programming to find the output $x_r$ of the $r$-th layer can be written as follows:

$$A_r x_r = b_r \tag{9}$$

where $A_r$ and $b_r$ is a matrix and vector depending on the previously reconstructed $x_{r+1}$, the parameter $\Theta_r$ of the $r$-the layer and its gradients. For the definition of $A_r$ and $b_r$, we refer to [32]. Since each linear programming has a closed-form solution $A_r^\dagger b_r$, this approach can be sometimes useful in terms of reducing computational cost.

**R-GAP with generative model.** Note that the problem in (9) can be rewritten as follows:

$$\min_{x_r} \|A_r x_r - b_r\| . \tag{10}$$

Let $f_{\theta,r}(x)$ be the output of the $r$-th layer. Then, we can interpret $f_{\theta,r}(G_w(z))$ as a generative model for $x_r$. Hence, the recursive reconstruction can be partially or fully replaced with the following optimization:

$$\min_{z,w} \|A_r f_{\theta,r}(G_w(z)) - b_r\| \tag{11}$$

where the search space can be alternated arbitrarily.

**A limitation of R-GAP.** Such a use of generative model in (11) provides the same gain from reducing searching space. We however want to note that it inherits the limitation of R-GAP, in which the reconstruction error in upper layers propagates to that in lower layer. Hence, as the learning model becomes deeper, the reconstruction quality decreases while the number of parameters is increasing. Figure A2 shows the phenomenon of error accumulation of R-GAP. It is possible that the optimization method in (11) can have lager error than the closed-form solution $A_r^\dagger b_r$ due to imperfection in generative model. Therefore, it is better to not use the generative model when the original linear programming is over-determined or determined. Indeed, in Figure A1, we present a trade-off between the linear programming in (11) and the optimization with generative model in (11), in which to emphasize the trade-off, we perform the latent space search over $z$ only. We obtain a substantial gain from using the generative model for a few layers (one or two), whereas the gradient inversion is failed when using the generative model for every layer.

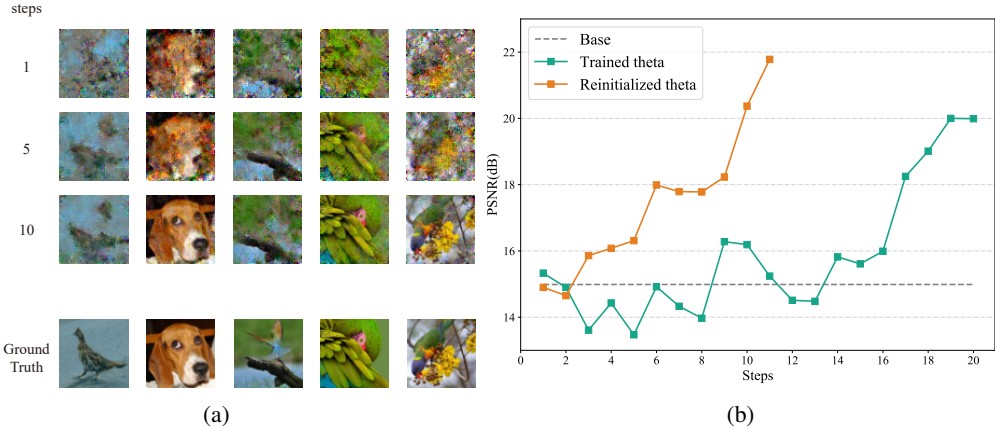

|  (a)  |  (b)  |

Figure A3: (a) Examples of reconstructed images with a sequence of gradients. A local dataset contains eight images and generates gradients using four randomly chosen images in every step. As time $t$ increases, reconstructed images become more accurate. (b) PSNR of reconstructed images. In reinitialized theta setting, the classification model is reinitialized every step. In trained theta setting, the classification model is trained every step.

# D    Another potential gain from inverting a set of gradients

In the main text, we demonstrate that from multiple gradients, we can train a generative model and use it to break the fundamental limit of inverting gradient solely. Beside this, assuming that we can observe a large number of gradients for the same data but different model parameters, it is able to reconstruct data almost perfectly by solving $\min_x \sum_{t=1}^T c(x; \theta_t, g_t)$. Such an assumption may be valid once we obtain the meta information to match gradients and data to be reconstructed. In Figure A3, we demonstrate this potential gain when there are eight images only, but we observe a sequence of gradients obtained from the procedure of FL (the green curve). Of course, in the procedure of FL, the model parameter $\theta$ slowly changes and thus the gain is smaller than that when each gradient is computed at completely random model parameters. However, in both settings, the reconstruction eventually becomes perfect as the observed gradients are accumulated.

# E    Strong generative prior

When we have stronger prior on the data distribution, the gain from the generative model becomes larger. To show this, we use FFHQ [12] rather than ImageNet in Section 5.1, where we believe FFHQ containing human-face images has less diversity than ImageNet including images of one thousand classes. With FFHQ data, even GI-$z$ significantly outperforms GI-$x$, while the gap between GI-$z$ and GI-$x$ is small for ImageNet in Figure 4b. This suggests a new approach to use a conditional generative model and data label $y^*$ in order for enjoying the gain from narrowing down the set of candidate input data by conditioning the label.

# F    Possible defense methods against gradient inversion attacks

In this section, we briefly discuss several defense algorithms against gradient inversion attacks.

**Disguising label information.**    As a defense mechanism specialized for the gradient inversion with generative model, we suggest to focus on mechanisms confusing the label reconstruction. In Section 5.1 and E, we observe that revealing the data label can curtail the candidate set of input data and thus provide a significant gain in gradient inversion by using conditional generative model. Therefore, by making the label restoration challenging, the gain from generative model may be decreasing. To be specific, we can consider letting node sample mini-batch to contain data having a certain number of labels, less than the number of data but not too small. By doing this, the possible combinations of labels per data in a batch increases and thus the labels are hard to recovered.

**Using large mini-batch.**    There have been proposed several defense methods against gradient inversion attacks [17, 16, 28], which let the gradient contain only small amount of information per data. Once a gradient is computed from a large batch of data, the quality of the reconstructed data using gradient inversion attacks fall off significantly, including ours(GI-$z/w$) as shown in Figure 5. The performance (PSNR) of GI-$z/w$, GI-$x$, and GI-$x$+BN are degenerated as batch size grows. However, we found that the degree of degeneration in GI-$x$+BN is particularly greater than that in ours, and from batch size 32, the advantage of utilizing BN statistics almost disappeared. This is because the benefit of BN statistics is divided by the batch size while the generative model helps each reconstruction in batch individually.

**Adding gaussian noise to the gradient.**    From the perspective of the differential privacy, adding noise to the gradient can prevent gradient inversion attacks from optimizing its objective. In our experiments, adding sufficiently large gaussian noise were able to prevent gradient inversion algorithms, including ours. [34] also provided a similar observation. Furthermore, we investigated that using large mini-batch with adding noise leverages the degree of degeneration. The result is shown in Figure 5. This also implies that one needs to add a larger noise when using a smaller batch size. In addition, this justifies employing a mechanism of secure multi-party computation with zero-sum antiparticles [23] or zero-mean noises [18] against our attack method. However, such a mechanism may increase implementation complexity or learning instability.

Such approaches can easily make the model training unstable. In general, we need to find a good balance in the trade-off between the stability of FL and the privacy leakage, while each defense mechanism has distinguishing pros and cons.

# G Convergence speed comparison with GIML

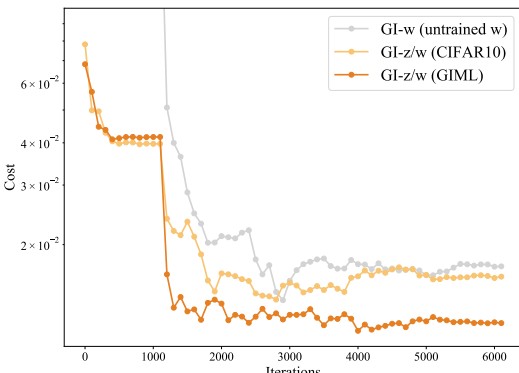

Figure A4: Gain of using generative model trained with GIML. A typical loss curve of reconstruction process. Meta-learned model converges faster than other model's results. Note that GI-$w$ does not perform latent space search of 0 to 1000 iterations.

Meta learning algorithms such as MAML [8] and Reptile [20] are often regarded as finding a good initialization for multiple tasks. In our case, each GIAS corresponds to a task. Thus, not only the performance of GIAS, the convergence speed of GIAS also increases. In Figure A4, we compare convergence speeds of GIAS with a meta-learned generative model, an wrong generative model(GIML), and an untrained generative model. The result shows using GIML also boost up the convergence speed of GIAS.

# H Experiment settings

Unless stated otherwise, we consider an image classification task on the validation set of ImageNet [22] dataset resized to $64 \times 64$ using a randomly initialized ResNet18 [10] as learning model. The resizing is necessary for computational tractability. Recalling the under-determined issue is the major challenge in gradient inversion, deeper and wider $f_\theta$ makes the gradient inversion easier [9]. Hence, the choice of ResNet18 as learning model is the most difficult setting within ResNet architectures since it contains the least information. Considering a trained ResNet as learning model results in slight drop of quantitative performance and large variance. We use a StyleGAN2[13] model trained on ImageNet for GIAS, in which the latent space search over $z$ implies the search over the intermediate latent space, known as $\mathcal{W}$ in the original paper [12], to improve the reconstruction performance, c.f., [1]. We use the batch size $B = 4$ as default, and negative cosine for the choice of gradient dissimilarity function $d(\cdot, \cdot)$, which apparently provides better inversion performance than $\ell_2$-distance in general [9]. For the optimization in GIAS, we use Adam optimizer [14] which decays learning rate by a factor of $0.1$ at $3/8, 5/8, 7/8$ of total iterations. from initial learning rates $\eta_z = 3 \times 10^{-2}$ for the latent space search and $\eta_w = 10^{-3}$ for the parameter space search. Since our experiments are conducted with image data, we used total variation regularizer with weight $\lambda_{\text{TV}} = 10^{-4}$ for all experiments. For each inversion, we pick the best recovery among $4$ random instances based on the final loss. All experiments are performed on GPU servers equipped with NVIDIA RTX 3090 GPU and NVIDIA RTX 2080 Ti GPU. Numerical results including graphs and table are averaged over 10 samples except Figure 3 and Figure A4.

Note that our baseline implementation for [29] includes fidelity regularizer with $\text{BN}_{\text{exact}}$ and group lazy regularizer, not group registration regularizer. Our baseline implementation might be imperfect, but it still demonstrates adding our method improves performance.

# I License of assets

**Dataset.** ImageNet data are distributed under licenses which allow free use for non-commercial research. FFHQ data are distributed under licenses which allow free use, redistribution, and adaptation for non-commercial purposes. ffhq-features-dataset provides annotations of FFHQ images. Original authors of FFHQ images are indicated in the metadata, if required. We did not include, redistribute, or change the data itself, and cited above three works. Note that there are still some concerns related to whether the data owners of the original images or the people within the images provided informed consent for research use.

**Source code.** Some parts of our source code came from open-source codes of several previous researches. For more details, see README of our source code.