# OpenReview forum: "Gradient Inversion with Generative Image Prior"
_NeurIPS.cc/2021/Conference — NeurIPS 2021 Poster_

### Official Review · Reviewer_A6z3 · 2021-07-12

**Rating:** 6
**Confidence:** 3

**Summary:**

The contributions of this paper are mainly two-fold:
1) Gradient inversion in alternative spaces (GIAS) that improves reconstruction with the help of the prior from generative models;
2) Gradient inversion to meta-learn (GIML) which learns a generative model via inverting multiple gradients computed on the data.
Via experiment results, this paper shows that GIAS with pretrained generative models can result in privacy leakage, and GIML can learn generative models from only gradients.

**Ethics Review Area:**

["Inappropriate Potential Applications & Impact  (e.g., human rights concerns)", "Privacy and Security (e.g., consent)"]

**Limitations And Societal Impact:**

As the authors acknowledged in the section of "societal impact and limitation", this paper mainly focuses on how using generative model prior can facilitate gradient inversion to reveal data. It would nice (and also necessary to some extent) to discuss possible defensive strategies especially against the methods proposed in this paper.

**Main Review:**

This paper proposes GIAS and GIML with improvements from baseline methods, while it might be better to provide more support on their effectiveness.

This paper compares the proposed GIAS with several other methods thoroughly via experiments. While I wonder if there are more theoretical justifications behind the effectiveness of the proposed searching mechanism in the latent and model parameter spaces.

On the other hand, how is the generalizability of GIML for more complex tasks? As 32x32 qualitative examples shown in Figure 4, there are substantial gaps between the reconstructions and the ground truths.



**Needs Ethics Review:**

Yes

**Time Spent Reviewing:**

3

---

> ### Author Response · Authors · 2021-08-11
> **Official Response to Reviewer A6z3**
>
> We sincerely thank you for the thorough review and the constructive and encouraging comments. We will revise the manuscript based on your comments.
>
> ## Comment 1
>
> This paper proposes GIAS and GIML with improvements from baseline methods, while it might be better to provide more support on their effectiveness. This paper compares the proposed GIAS with several other methods thoroughly via experiments. While I wonder if there are more theoretical justifications behind the effectiveness of the proposed searching mechanism in the latent and model parameter spaces.
>
> ## Response 1
>
> > Our approach, in fact, builds upon several techniques in machine learning, including compressed sensing with generative priors [Bora’17] (the latent space search in GIAS), Deep Image Prior [22] (the parameter space search in GIAS), and meta-learning for representation learning [Tripuraneni’20] (GIML), which are theoretically understood at varying depths. For instance, when reconstructing an image from compressed gradient, the latent space search in GIAS corresponds to the compressed sensing with generative priors [Bora’17]. It is theoretically well understood that there is a significant gain in using a trained generative model. In short, the dimension of the observations required to solve the inverse problem scales as the dimension of an image $m$ when we do not use a trained generative model, but this reduces to the dimension of the latent code $k$ when we use a trained generative model. As typically $k<<m$, this and Property 1 justify the gain of using a well trained generative model. In addition, the parameter space search (using a generative model consisting of CNN for image reconstruction), followed after the latent space search, can be theoretically supported by the theory explaining deep image prior [22], e.g., implicit bias [Gunasekar’18]. Finally, while a rigorous theoretical analysis of GIML is outside the scope of this paper, the use of meta-learning in this context is justified by recent theoretical advances in meta-learning. It is shown in [Tripuraneni’20] that each task can be solved to a significantly improved accuracy (which corresponds to recovering each image from its gradient for our setting) by leveraging a common shared representation (which corresponds to a common generative model in our case) over several tasks.
>
>
> ---
>
> ## Comment 2
> On the other hand, how is the generalizability of GIML for more complex tasks? As 32x32 qualitative examples shown in Figure 4, there are substantial gaps between the reconstructions and the ground truths.
>
> ## Response 2
>
> > We note that Figure 4b reports the reconstruction result when observing only 5% of entries of gradients, i.e., the sparsity is 0.95 and thus this is a fairly complex and challenging task. In an easy scenario (e.g., no sparsity), of course, we can close the gap between the reconstructions and the ground truths, but there will be no significant difference between the reconstruction methods. We report the results in the challenging task with sparsity 0.95 to show the gain of meta-learning generative model compared to using wrong prior (CIFAR10).
>
> ---
>
> ## Comment 3
>
> As the authors acknowledged in the section of "societal impact and limitation", this paper mainly focuses on how using generative model prior can facilitate gradient inversion to reveal data. It would be nice (and also necessary to some extent) to discuss possible defensive strategies especially against the methods proposed in this paper.
>
> ## Response 3
>
> > Our major contribution is to give a warning message to the FL society by emphasizing that gradient based attacks can break down defense methods which were safe according to the previous standard. Since the typical defense strategies (increasing mini-batch size, adding noise, sparsifying or compressing gradient and FedAvg) are still effective to our method, we can use them but with more conservative choices of hyperparameters than before. As described in Appendix E, we have a rough idea of defensive strategy specialized for our attack with generative priors. Our method becomes more effective when the conditional distribution of data given the generative model and gradient has smaller entropy. Hence, composing a mini-batch to maximize the entropy of labels so that to make label reconstruction difficult can be a specialized defensive mechanism.
>
> ---
> ## References
> [22] Deep Image Prior, D Ulyanov, A Vedaldi, V Lempitsky
>
> [Bora’17] Compressed sensing using generative models, A Bora, A Jalal, E Price, AG Dimakis.
>
> [Tripuraneni’20] Provable meta-learning of linear representations, N Tripuraneni, C Jin, M Jordan
>
> [Gunasekar’18] Implicit Bias of Gradient Descent on Linear Convolutional Networks, S Gunasekar, J Lee, D Soudry, N Srebro

---

### Official Review · Reviewer_6jyM · 2021-07-15

**Rating:** 7
**Confidence:** 5

**Summary:**

The submission "Gradient Inversion with Generative Image Prior" investigates privacy attacks against federated learning which invert gradient information sent by users to uncover private information. They authors propose generative models as tools to improve inversion, utilizing both optimization in latent space and parameter finetuning of such generative models. This approach improves the success rates of gradient inversion attacks against image classification (and especially best-case PSNR).

**Limitations And Societal Impact:**

The authors propose an improved attack against privacy in federated learning. They appropriately discuss the societal impact of this measure in a separate subsection.

**Main Review:**

Overall I think this work is a reasonable improvement of gradient inversion attacks against federated learning. The idea to utilize prior knowledge via generative models is good and executed well. I have a several main comments which I would like the authors to consider:

Comments:
* I would like to see a larger discussion of [23] (Wang et al., Beyond Inferring Class Representatives) in the related work section, which is an older work in the gradient inversion literature, but also involves training a GAN to aid in the reconstruction (although not quite in the way the authors describe here, but it would be good to delineate the differences of both approaches explicitly)
To a lesser degree this comment also applies to works such as [Hitaj et al, Deep Models Under the GAN: Information Leakage from Collaborative Deep Learning]
* In [9] (Geiping et al, Inverting Gradients), small prior effects are noticed when considering GI-x reconstruction from the gradients of trained models. For the given submission I am not quite clear whether the authors consider a randomly initialized ResNet-18 model are a trained model. In any case, does the beneficial effect of the generative model increase or decrease when switching from random to trained parameters?
* Feedback from practitioners of federated learning about gradient attacks has often focussed on the perceived security of larger image batches (via e.g secure aggregation) and gradient noise (as main component of differential privacy). It would be great if the authors could plot the success of the proposed reconstruction approach (possibly in the best-PSNR case) over a large range of batch sizes (up to 48 or even 128?) and gradient noise levels. This would be an extension of Fig.3. While there is of course the expectation that all attacks fall off significantly with larger batch sizes, the relative gains in falloff over previous attacks can be instructive for many readers (especially if the authors could show that their method has a noticeably slower falloff rate than others, which is difficult to make out from Fig.3 currently). Alternatively a measure of success along the lines of Fig.8 in [25] (Yin et al, See through gradients) could be investigated.
* If I understand l236 correctly then this submission does not investigate a comparison to the full attack scheme of [25]? This is understandable, given the recency of that work, but it would be great if the authors could ultimately provide a clearer comparison between both attack schemes to confirm that the improvements of the generative model stack even with the full attack of [25]


Minor comments that do not influence my assessment:
* I think the authors could title this paper "Gradient Inversion with Generative Image Priors" (note the added _s_). The usage of "generative image prior" in the singular feels a bit off to me grammatically and further brings up associations with works such as "Deep image prior" and its successors, even though a deep image prior is not "the prior" used in this work (as StyleGAN is used). This could be avoided with a different title.
* The paper could benefit from another round of proof-reading by the authors to iron out minor grammatical inconsistencies.

**Time Spent Reviewing:**

2.5

---

> ### Author Response · Authors · 2021-08-11
> **Official Response to Reviewer 6jyM**
>
> We sincerely thank you for the thorough review and the constructive and encouraging comments. We will revise the manuscript based on your comments.
>
> ## Comment 1
>
> I would like to see a larger discussion of [23] (Wang et al., Beyond Inferring Class Representatives) in the related work section, which is an older work in the gradient inversion literature, but also involves training a GAN to aid in the reconstruction (although not quite in the way the authors describe here, but it would be good to delineate the differences of both approaches explicitly) To a lesser degree this comment also applies to works such as [Hitaj et al, Deep Models Under the GAN: Information Leakage from Collaborative Deep Learning].
>
> ## Response 1
>
> > We will include the following discussion on the [23] and [Hitaj’17] in our revision. As you mentioned, those two works are indeed related to ours as they also train a generative model in FL to demonstrate privacy leakage. However, there are two clear differences compared to our work. First, in our work, we not only train a generative model but also utilize it for reconstruction, while [23] and [Hitaj’17] do not use the generative model for the reconstruction. Hence, In our approach, the generative model and reconstruction can be improved interactively to each other as shown in Figures 4a and 5a. Second, [23] and [Hitaj’17] require some auxiliary dataset given in advance to enable the training of GAN, but we train a generative model using gradients only. In addition, [23] is less sample-efficient than ours in a sense that they use gradients to reconstruct images and then train a generative model with the reconstructed images, i.e., if the reconstruction fails, then the corresponding update of the generative model fails too, whereas we train the generative model directly from gradients.
>
> ---
>
> ## Comment 2
>
> In [9] (Geiping et al, Inverting Gradients), small prior effects are noticed when considering GI-x reconstruction from the gradients of trained models. For the given submission I am not quite clear whether the authors consider a randomly initialized ResNet-18 model are a trained model. In any case, does the beneficial effect of the generative model increase or decrease when switching from random to trained parameters?
>
> ## Response 2
>
> > In our main experiments, we had used a randomly initialized ResNet-18 classifier. Since the reviews were released, we have been running a set of experiments using a trained classifier. As Geiping [9] reported, our method to invert gradients based on the trained classifier shows a slight drop of average performance in PSNR (-3.5), SSIM (-0.05), and LPIPS (+0.025). We however found a new interesting tendency of our method that the best-in-batch PSNR is improved despite the drop of average PSNR as shown in the batch reconstruction result at link (https://imgur.com/dLejbRT). Hence, in terms of the best-in-batch PSNR, the benefit of using the generative model with the trained classifier is larger than that with the randomly initialized classifier. We will look into this phenomenon thoroughly with more experiments and share as soon as possible.
>
> ---
>
> ## Comment 3
>
> Feedback from practitioners of federated learning about gradient attacks has often focussed on the perceived security of larger image batches (via e.g secure aggregation) and gradient noise (as main component of differential privacy). It would be great if the authors could plot the success of the proposed reconstruction approach (possibly in the best-PSNR case) over a large range of batch sizes (up to 48 or even 128?) and gradient noise levels. This would be an extension of Fig.3. While there is of course the expectation that all attacks fall off significantly with larger batch sizes, the relative gains in falloff over previous attacks can be instructive for many readers (especially if the authors could show that their method has a noticeably slower falloff rate than others, which is difficult to make out from Fig.3 currently). Alternatively a measure of success along the lines of Fig.8 in [25] (Yin et al, See through gradients) could be investigated.
>
> ## Response 3
>
> > We experimented with batch size up to 48. We provide the corresponding plot in this link (https://imgur.com/F9oLMMO), where the PSNR of GIAS (GI-$z/w$) at batch size 16, 32, 48 were 14.902, 14.079 and 13.760. It is a 7%, 12% and 14% drop for each compared to batch size 4 (PSNR 16.055). The PSNR of GI-$x$ + BN at batch size 16, 32, 48 were 13.303, 12.097 and 11.675. It is a 14%, 22%, and 24% drop for each compared to batch size 4 (PSNR 15.447).
>  As expected from the earlier works, the performance (PSNR) of our method (GI-$z/w$) and the others (GI-$x$ [9] and GI-$x$+BN [25]) are degenerated as batch size grows. However, we found that the degree of degeneration in GI-$x$+BN [25] is particularly greater than that in ours, and from batch size 32, the advantage of utilizing BN statistics almost disappeared. This is because the benefit of BN statistics is divided by the batch size while the generative model helps each reconstruction in batch individually.
>
> > The experiments about adding a various level of noise to gradients is now in progress, and we will share the result as soon as possible. We have tried to evaluate using IIP (the measure that you mentioned) in [25] as we agree with you. However, it is unfortunate that we were not able to obtain the authors’ implementation of IIP and there are some difficulties in implementing it ourselves due to some unclear description.  Hence, we would leave the investigation about a new measurement of reconstruction success as a future work.
>
> ---
>
> ## Comment 4
>
> If I understand l236 correctly then this submission does not investigate a comparison to the full attack scheme of [25]? This is understandable, given the recency of that work, but it would be great if the authors could ultimately provide a clearer comparison between both attack schemes to confirm that the improvements of the generative model stack even with the full attack of [25].
>
> ## Response 4
>
> > We agree that we had to compare a full attack scheme of [25] to get a clearer comparison. However, the official implementation of [25] is not available as of now, and it was unfortunate that the authors of [25] were not able to share their code nor implementation details when we requested. We hence implemented and compared the methods of [25] as many as we can (BN statistics and group regularization). We believe that using the generative model would provide an independent gain in addition to the full attack scheme as it did with BN statistics.
>
> ---
>
> ## Comment 5
>
> Minor comments that do not influence my assessment:
> I think the authors could title this paper "Gradient Inversion with Generative Image Priors" (note the added s). The usage of "generative image prior" in the singular feels a bit off to me grammatically and further brings up associations with works such as "Deep image prior" and its successors, even though a deep image prior is not "the prior" used in this work (as StyleGAN is used). This could be avoided with a different title.
> The paper could benefit from another round of proof-reading by the authors to iron out minor grammatical inconsistencies.
>
> ## Response 5
> > We thank you for the detailed editorial comments. We will revise the manuscript accordingly. In particular, we feel the same about the current title, and we will consider changing the title as you suggested.
>
> ---
> ## References
>
> [9] Inverting gradients–How easy is it to break privacy in federated learning?, J. Geiping, H. Bauermeister, H. Dröge, and M. Moeller.
>
> [23] Beyond inferring class representatives: User-level privacy leakage from federated learning., Z. Wang, M. Song, Z. Zhang, Y. Song, Q. Wang, and H. Qi.
>
> [25] See through gradients: Image batch recovery via gradinversion., H. Yin, A. Mallya, A. Vahdat, J. M. Alvarez, J. Kautz, and P. Molchanov.
>
> [Hitaj’17] Deep models under the GAN: information leakage from collaborative deep learning, B Hitaj, G Ateniese, F Perez-Cruz

---

> > ### Comment · Reviewer_6jyM · 2021-08-23
> > **Thank you for the clarification.**
> >
> > Thank you for providing detailed feedback for my questions. I don't have further questions and have decided to increase my score to 7.

---

> ### Author Response · Authors · 2021-08-22
> **Response 3-1 to Reviewer 6jyM**
>
> ## Response 3-1
>
>
> > The experiment with varying additive noise to gradients has been completed. We provide the result in [exp-noise](https://imgur.com/9u5dQP9). Adding sufficiently large noise (> $10^{-1}$ with batch size 4 or  $10^{-2}$ with batch size 32) were able to prevent gradient inversion attacks, including ours. This also implies that one needs to add a larger noise when using a smaller batch size. A similar observation is provided in [31]. In addition, this justifies employing a mechanism of secure multi-party computation with zero-sum antiparticles [Bonawitz’17] or zero-mean noises [16] against our attack method. However, such a mechanism may increase implementation complexity or learning instability. In the revision, we will include the above experiment and discussion to provide ideas to defend against our attack method and others.
>
> ## References
>
> [16] Learning differentially private recurrent language models, H. B. McMahan, D. Ramage, K. Talwar, and L. Zhang.
>
> [31] Deep leakage from gradients, L. Zhu et al.
>
> [Bonawitz’17] Practical secure aggregation for privacy-preserving machine learning, Bonawitz, Keith, et al.

---

### Official Review · Reviewer_oRD2 · 2021-07-16

**Rating:** 6
**Confidence:** 4

**Summary:**

This paper proposed training a generative model for gradient inversion. Specifically, the authors proposed learning generative models by leveraging a generative image prior and learn a generative model for each data within one gradient inversion task. Empirical results show that the proposed GIAS and GIML outperforms baselines in applications of federated learning.

**Limitations And Societal Impact:**

Limitations and suggestions are listed in the main review.

**Main Review:**

The paper proposed a novel method to optimize objective (3) using a generative model by searching latent space and parameter space. In Algorithm 1, the author proposed first searching the latent space then searching the parameter space. However, it is not clear to me why it is not done in an alternate fashion. Firstly, the authors seem to decompose (3) into (6) and (8). With algorithm 1, only (8) is satisfied. There's the z_i's solved by (6) no longer minimize (3) given the solutions in (8). Clarification of why algorithm 1 is the desired method should be stated here.

The proposed GIML algorithm learns a generative model. In the experiment, a DCGAN is learnt by the proposed algorithm. However, learning a GAN requires real image data. What is used as the real image data here? In the scenario of federated learning, who is doing the gradient inversion? The server or other clients. If the server is inverting the gradient, does that mean the server need to store a public dataset. Moreover, if the public dataset distribution is highly heterogeneous with the local data distribution, what would the performance be. In the experiment section, the heterogeneity of client data distribution is not discussed. I think it would significantly improve the experiment results if the relation between performance and heterogeneity level is presented.

**Time Spent Reviewing:**

3

---

> ### Author Response · Authors · 2021-08-11
> **Official Response to Reviewer oRD2**
>
> We sincerely thank you for the thorough review and all the constructive/editorial comments. We will revise the manuscript based on your comments.
>
> ## Comment 1
>
> The paper proposed a novel method to optimize objective (3) using a generative model by searching latent space and parameter space. In Algorithm 1, the author proposed first searching the latent space then searching the parameter space. However, it is not clear to me why it is not done in an alternate fashion. Firstly, the authors seem to decompose (3) into (6) and (8). With algorithm 1, only (8) is satisfied. There's the z_i's solved by (6) no longer minimize (3) given the solutions in (8). Clarification of why algorithm 1 is the desired method should be stated here.
>
> ## Response 1
>
> > The idea optimizing $z$ followed by optimizing $w$ was inherited from [3]. With a pretrained generative model, the solution of (6) finds overall structures and the next step (optimizing $w$) finds fine details. In Figures 1, 2 and A4, we provide numerical justification showing that optimizing either $z$ or $w$ is insufficient. In addition, we had also considered the other design choices such as jointly training $z$ and $w$ (i.e., GI-$(z,w)$) or iteratively alternating $z$ and $w$ (i.e., GI-$z/w/z/w/...$), which did not provide any further gain despite the additional computational cost compared to GI-$z/w$. We hence propose GI-$z/w$. We will add further discussion on the rationale of GI-$z/w$ in our revision.
>
> ---
>
> ## Comment 2
>
> The proposed GIML algorithm learns a generative model. In the experiment, a DCGAN is learnt by the proposed algorithm. However, learning a GAN requires real image data. What is used as the real image data here?
>
> ## Response 2
>
> > To clarify, GIML does not require any real image data since it trains only the generator of DCGAN but no discriminator. Hence, our method is applicable even when an auxiliary dataset of real data is unavailable.
>
> ---
>
> ## Comment 3
>
> In the scenario of federated learning, who is doing the gradient inversion? The server or other clients. If the server is inverting the gradient, does that mean the server need to store a public dataset?
>
> ## Response 3
>
> > We basically consider the scenario of a server reconstructing data from gradients collected from nodes, also known as the honest-but-curious server setting [9]. We note that since a participating node also has access to the classifier model parameter and the aggregated gradient, one can think of a node performing the gradient inversion although it may not know which reconstructed data belongs to which node. We again note that as we demonstrated in Figure 4 and 5, our approach (GIML) does not require the server to store any public dataset, although it may be helpful.
>
> ---
>
> ## Comment 4
>
> Moreover, if the public dataset distribution is highly heterogeneous with the local data distribution, what would the performance be? In the experiment section, the heterogeneity of client data distribution is not discussed. I think it would significantly improve the experiment results if the relation between performance and heterogeneity level is presented.
>
> ## Response 4
>
> > As you mentioned, learning and using a narrower distribution can be more effective. In case of the highly heterogeneous dataset, we can perform GIML per node for given a sequence of gradients over time. By doing this, we can narrow down the target distribution learned by GIML from the heterogeneous one (e.g., human faces) to the small one (e.g., old male with eyeglasses), and thus we can expect substantial benefit in reconstructing input data. We will definitely try to include an experiment showing this observation in our revision.
>
> ---
>
> ## References
>
> [3] Semantic photo manipulation with a generative image prior., D. Bau, H. Strobelt, W. Peebles, J. Wulff, B. Zhou, J.-Y. Zhu, and A. Torralba.
>
> [9] Inverting gradients–How easy is it to break privacy in federated learning?, J. Geiping, H. Bauermeister, H. Dröge, and M. Moeller.

---

### Official Review · Reviewer_ofcZ · 2021-07-19

**Rating:** 6
**Confidence:** 3

**Summary:**

The paper considers the interesting Federated Learning problem of recovering latent code data using gradient information, in the presence of additional prior knowledge and without. For an unknown dataset $\{(x_i,y_i)\}_{i=1}^B$, where $y_i$ is the label of $x_i$, the task is to find $x_i$ given gradient of $l(f_\theta(x_i),y_i)$ with respect to $\theta$. Here, $f_\theta$, is a classifier with parameters $\theta$ and $l$ is a appropriate loss function.

In the case where a generative prior on the dataset $\{x_i\}$ is known, the paper introduces the Gradient Inversion in Alternate Spaces (GIAS) algorithm that searches the latent code space of the generator to match the known gradients. In the case where a generative prior is not known, the paper introduces the Gradient Inversion to Meta-Learn generative model (GIML) algorithm, which, in two stages, learns weights of a generative model and searches the latent code case of the approximate generative model to match the gradients (up to a regularization term). The paper is primarily focused on the implementation of these algorithms. Experimental results were provided to show the efficiency of the proposed algorithms on real datasets.

**Limitations And Societal Impact:**

The paper discusses its limitation in the conclusion.

**Main Review:**

The paper provides a sufficient description of prior work. However, some sentences used in the paper are not clear or have errors. For example:
1. Lines 53-55
2. Line 66: propose should be proposed
3. Is BN in Line 73 Batch Norm? This needs to be made clear.
4. Lines 75-77
5. Lines 94-96
6. Lines 265-266

To my knowledge, the paper assumes that the classifier $f_\theta$ is known (or at least the architecture on $f_\theta$ is known). Is this a valid and reasonable assumption in practice?

Is there anything we can say about the number of gradients we need to accurately recover/estimate the dataset. For the case when the generative model is not known, is it in the order of $k$ (the latent code dimension)? What about when the generative model is known? A discussion would be helpful.

What is the motivation for line 11 in Algorithm 2.

In practice, how does one assume the architecture of the generative model if such a model is not known a prior?

**Time Spent Reviewing:**

5

---

> ### Author Response · Authors · 2021-08-11
> **Official Response to Reviewer ofcZ**
>
> We sincerely thank you for the thorough review and the constructive/editorial comments. We will revise the manuscript based on your comments.
>
>
> ## Comment 1
>
> To my knowledge, the paper assumes that the classifier $f_\theta$ is known (or at least the architecture on $f_\theta$ is known). Is this a valid and reasonable assumption in practice?
>
> ## Response 1
>
> > As you understood and every prior work [9, 25, 27, 30] did, we consider the privacy leakage when the attacker has access to $f_\theta$ and gradients (or model updates). In the standard federated learning, the central server distributes $f_\theta$ to nodes and collects node gradients computed from $f_\theta$. Hence, it can perform the gradient inversion to reconstruct each node’s data. In addition, since a participating node knows $f_\theta$ and the update of $f_\theta$, it can reconstruct the others’ data although it may not know which reconstructed data belongs to which node.
>
> ---
>
> ## Comment 2
>
> Is there anything we can say about the number of gradients we need to accurately recover/estimate the dataset. For the case when the generative model is not known, is it in the order of $k$ (the latent code dimension)? What about when the generative model is known? A discussion would be helpful.
>
> ## Response 2
>
> > Even though a rigorous analysis is outside the scope of this paper, we provide a rough conjecture about the sample complexity in terms of the number of gradients for reconstructing data and learning a generative model. First, suppose that the generative model of $k$-dimensional latent space is known, each gradient is computed from subsampling a batch of size $B$. Let $g$ be the number of gradients we observe, and $N$ be the dimension of the gradient. We are trying to recover $Bk$ scalar variables with $gN$ scalar observations. We would expect that we need $g>Bk/N$ to be able to recover the images assuming that we know whether each data is used to compute a gradient or not and each gradient is computed at fairly different points. In Appendix C and Figure A3, we report an experiment on the sample complexity lower bound.
>
> > When a generative model is unknown, and to obtain that generative model with GIML, we need to meta-learn from multiple tasks jointly. Again, although a rigorous analysis is outside the scope of this paper, we follow Theorem1 of [Tripuraneni’20] to get an intuition on how many samples might be needed. We are jointly learning from $T$ tasks, where each task is to find a mini-batch of $B$ images from $g$ gradient observations. Let $M$ denote the dimension of the parameters for the generative model. Following the analysis of [Tripuraneni’20], we need $g>M/(NT)$ and $g>Bk/N$. The first condition ensures that the total dimension of observations ($T$ tasks, $g$ gradients each, and $N$ dimension for each gradient) is larger than the size of the unknown parameter of the generative model. The second condition ensures that the number of observations per task is larger than the dimension of the task specific unknowns.
>
> ---
>
> ## Comment 3
>
> What is the motivation for line 11 in Algorithm 2?
>
> ## Response 3
>
> > The hyperparameter $\beta$ in line 11 is employed to slow down the timescale of meta update compared to that of individual update (governed by $\tau$ and $\alpha$). From this, we can prevent overfitting to few inversion tasks. Such a technique in fact originated from the gradient-based meta learning algorithms such as MAML [8] and Reptile [Nichol’18].
>
> ---
>
> ## Comment 4
>
> In practice, how does one assume the architecture of the generative model if such a model is not known a prior?
>
> ## Response 4
>
> > Our methods, GIAS and GIML, only assume a canonical form of generative model that is a mapping from simple/low-dimensional latent vector to data. Hence, for any typical type of data (image, audio or language), we can easily find a good enough architecture among the off-the-shelf ones having the canonical form. Of course, if we have more information on the dataset, it may be possible to employ a specialized architecture (or pretrained model) for easier training of the generative model and inverting gradients. In case of dealing with an unusual type of data, one may design a generative model based on the inverse function of the feature extractor (which is typically a mapping from data to low-dimensional feature vector) in the classifier.
>
> ---
> ## References
>
> [8] Model-Agnostic Meta-Learning for Fast Adaptation of Deep Networks., C. Finn, P. Abbeel, and S. Levine.
>
> [9] Inverting gradients–How easy is it to break privacy in federated learning?, J. Geiping, H. Bauermeister, H. Dröge, and M. Moeller.
>
> [25] See through gradients: Image batch recovery via gradinversion., H. Yin, A. Mallya, A. Vahdat, J. M. Alvarez, J. Kautz, and P. Molchanov.
>
> [27] iDLG: Improved deep leakage from gradients., B. Zhao, K. R. Mopuri, and H. Bilen.
>
> [30] Deep leakage from gradients., L. Zhu, Z. Liu, and S. Han.
>
> [Tripuraneni’20] Provable meta-learning of linear representations, N Tripuraneni, C Jin, M Jordan
>
> [Nichol’18] On first-order meta-learning algorithms, A Nichol, J Achiam, J Schulman

---

### Review · Ethics_Reviewer_2Zj9 · 2021-08-11

**Recommendation:**

Ideally, the research would have a defense to this privacy attack available at the same time that the issue is demonstrated.  Short of that, a more robust discussion of _potential_ protection mechanisms, eg whether current methods such as DP might help, would be useful.

If this attack is demonstrated against a federated learning algorithm in broad use, and in particular a real-world deployed version of the algorithm, I would be very concerned that this work would be "give unbalanced power to malicious entities".  As is, I think the paper falls in a more gray area regarding the trade-off the authors mention.



**Ethical Issues:**

Yes

**Ethics Review:**

The core ethical issue at play is related to the methodological example presented in the NeurIPS ethics guidelines:

   > On the methodology side, for example, a new adversarial attack might give unbalanced power to malicious entities; in this case, defenses and other mitigation strategies would be expected, as is standard in computer security.

This paper presents an issue similar to the example above by identifying a method for performing a privacy attack in a federated learning architecture, without also identifying and validating a suitable defense.   The concern, of course, is that this demonstration provides unbalanced power to malicious entities.  While potential mitigations (such as use of differential privacy within the FL architecture) are available, they are not evaluated and it is unclear if they protect against this attack while also allowing learning to occur.

Separately, the paper's use of the datasets based on crawling the web and/or Flickr---and in particular images of people (ffhq) gathered in this way---raises some concerns related to consent and whether the data owners of the original crawled images and the people within the images provided informed consent for research use, even if the data is available via creative commons otherwise. [see e.g., https://arxiv.org/abs/2006.16923].  Standards on what is acceptable and what is not are evolving or unresolved, but worth mentioning.

---

> ### Author Response · Authors · 2021-08-22
> **Official Response to Ethics Reviewer 2Zj9**
>
> We sincerely thank you for the ethics review and comments. We will revise the manuscript based on your comments.
>
> ## Comment 1
>
> This paper presents an issue similar to the example above by identifying a method for performing a privacy attack in a federated learning architecture, without also identifying and validating a suitable defense. The concern, of course, is that this demonstration provides unbalanced power to malicious entities. While potential mitigations (such as use of differential privacy within the FL architecture) are available, they are not evaluated and it is unclear if they protect against this attack while also allowing learning to occur.
>
> ## Response 1
>
> > Our major contribution is to warn the FL community to use a higher standard on privacy defined by our attack mechanism, and to raise the necessity of a more conservative choice of defense mechanisms than before. In the following links ([exp-batch-size] (https://imgur.com/V77BhLA) and [exp-noise](https://imgur.com/9u5dQP9)), we provide experiment results (which includes additional ones after receiving review) demonstrating that against our attack method, existing defense mechanisms (using large batches, adding noise to gradients, compressing gradients etc.) can be applied, but we need much more conservative ones to meet the higher standard that we propose. First, [exp-batch-size](https://imgur.com/V77BhLA) shows that increasing batch size can provide privacy protection against inversion attacks including ours (GI-z/w). However, for a certain level of privacy (average PSNR < 13), our method (GI-z/w) suggests using batch size 48 or larger that is much higher than the previous standards (> 8: GI-x [9] or 16: GI-x + BN [25] with access to batch normalization (BN) statistics). We note that comparing GI-x and GI-x+BN, the additional threat from sharing BN statistics, is decreasing as the batch size increases, while that from the generative model is not. Hence, this tells us that even when not sharing BN statistics (c.f., FL with BN layers without BN stats [Li’21]), we need to be much more conservative than what we thought before. In [exp-noise](https://imgur.com/9u5dQP9), each attack method has reconstruction quality dropping rapidly at a certain level of noise (> $10^-1$ or $10^-2$). This implies that employing a secure multi-party computation with zero-sum antiparticles [Bonawitz’17] or zero-mean noises [16] can be considered a fundamental defense mechanism, although this may increase implementation complexity or learning instability. We will revise our manuscript including the above experiment and discussion to provide guidelines for protecting privacy against our attack method and others.
>
> ## Comment 2
>
> Separately, the paper's use of the datasets based on crawling the web and/or Flickr---and in particular images of people (ffhq) gathered in this way---raises some concerns related to consent and whether the data owners of the original crawled images and the people within the images provided informed consent for research use, even if the data is available via creative commons otherwise. [see e.g., https://arxiv.org/abs/2006.16923 ]. Standards on what is acceptable and what is not are evolving or unresolved, but worth mentioning.
>
>
> ## Response 2
>
> > Thanks for letting us know of this issue and the useful reference. We selected those human-face images to give a strong warning of privacy leakage. However, in the revision, we will find a replacement in quantitative evaluation. In addition, as you suggested, we will also include this discussion with proper references in the revision for future researchers.
>
>
> ## References
>
> [9] Inverting gradients–How easy is it to break privacy in federated learning?, J. Geiping, H. Bauermeister, H. Dröge, and M. Moeller.
>
> [16] Learning differentially private recurrent language models, H. B. McMahan, D. Ramage, K. Talwar, and L. Zhang.
>
> [25] See through gradients: Image batch recovery via gradinversion., H. Yin, A. Mallya, A. Vahdat, J. M. Alvarez, J. Kautz, and P. Molchanov.
>
> [Li’21] FedBN: Federated Learning on Non-IID Features via Local Batch Normalization, Xiaoxiao Li, Meirui Jiang, Xiaofei Zhang, Michael Kamp, Qi Dou
>
> [Bonawitz’17] Practical secure aggregation for privacy-preserving machine learning, Bonawitz, Keith, et al.

---

> > ### Comment · Ethics_Reviewer_2Zj9 · 2021-08-29
> > **thank you for your response.  these changes will improve the ethical considerations of the paper**
> >
> > I thank the authors for their response.  The additions of an experiment to demonstrate a potential fundamental defense mechanism as guidance, and the promise to replace human-face images in the paper's quantitative evaluation are welcome revisions and will significantly improve the ethics aspects of the paper.

---

> ### Comment · Ethics_Reviewer_D9k8 · 2021-08-29
> **Thank you!**
>
> I thank the authors for this thoughtful response. You've addressed my concerns and I think the revisions to the paper will only strengthen it!

---

### Review · Ethics_Reviewer_D9k8 · 2021-08-14

**Recommendation:**

In many respects, this follows best practices in privacy and security research in other fields. Of course, the authors might have also reflected on whether there are any actors who might be especially vulnerable to immediate and effective attack and --- if such actors exists --- alerted them to these risks and give them a chance to address these risks prior to the publication of the paper. They could have also been more explicit about the way that they perform the cost benefit analysis. Presumably the authors are relying on certain expectations about the likelihood and severity of attacks and comparing these to how much their disclosure will spur useful change. Without a more explicit discussion of these expectations, it is impossible to evaluate the quality of their cost-benefit analysis.

**Ethical Issues:**

Yes

**Ethics Review:**

Like all work demonstrating the efficacy of new privacy attacks, this paper does, of course, raise some ethical concerns insofar as it provides a new method to potentially violate people's privacy. At the same time, there are benefits to disclosure if such disclosure helps draw attention to the vulnerabilities and spurs further research to develop effective mitigations. The challenge is to strike an appropriate balance.

---

> ### Author Response · Authors · 2021-08-22
> **Official Response to Ethics Reviewer D9k8**
>
> We sincerely thank you for the ethics review and comments. We will revise the manuscript based on your comments.
>
> ## Comment 1
>
> Of course, the authors might have also reflected on whether there are any actors who might be especially vulnerable to immediate and effective attack and --- if such actors exists --- alerted them to these risks and give them a chance to address these risks prior to the publication of the paper.
> They could have also been more explicit about the way that they perform the cost benefit analysis. Presumably the authors are relying on certain expectations about the likelihood and severity of attacks and comparing these to how much their disclosure will spur useful change. Without a more explicit discussion of these expectations, it is impossible to evaluate the quality of their cost-benefit analysis.
>
>
> ## Response 1
>
> > Our work brings attention to how effective gradient inversion attacks can be, thus tilting the scale towards more secure (but costly) federated learning protocols in the cost-benefit analysis. We hope this work justifies investing more resources to a more secure protocol for those FL systems running in practice. The cost of disclosure of our results is not expensive, as current defense mechanisms are still effective. To be specific, in the following links ([exp-batch-size](https://imgur.com/V77BhLA) and [exp-noise](https://imgur.com/9u5dQP9)), we provide experiment results (which includes additional ones after receiving review) demonstrating that against our attack method, existing defense mechanisms (using large batches, adding noise to gradients, compressing gradients etc.) can be applied, but we need much more conservative ones to meet the higher standard that we propose, although they may require some additional cost (e.g., increase implementation complexity or learning instability). We will include those results and discussions in the revision.
>
> > In what follows, we provide more detailed discussion on the additional experiments:
>
> > [exp-batch-size](https://imgur.com/V77BhLA) shows that increasing batch size can provide privacy protection against inversion attacks including ours (GI-z/w). However, for a certain level of privacy (average PSNR < 13), our method (GI-z/w) suggests using batch size 48 or larger that is much higher than the previous standards (> 8: GI-x [9] or 16: GI-x + BN [25] with access to batch normalization (BN) statistics). We note that comparing GI-x and GI-x+BN, the additional threat from sharing BN statistics, is decreasing as the batch size increases, while that from the generative model is not. Hence, this tells us that even when not sharing BN statistics (c.f., FL with BN layers without BN stats [Li’21]), we need to be much more conservative than what we thought before.
>
> > In [exp-noise](https://imgur.com/9u5dQP9), each attack method has reconstruction quality dropping rapidly at a certain level of noise (> $10^-1$ or $10^-2$). This implies that employing a secure multi-party computation with zero-sum antiparticles [Bonawitz’17] or zero-mean noises [16] can be considered a fundamental defense mechanism, although this may increase implementation complexity or learning instability.
>
>
> ## References
> [9] Inverting gradients–How easy is it to break privacy in federated learning?, J. Geiping, H. Bauermeister, H. Dröge, and M. Moeller.
>
> [16] Learning differentially private recurrent language models, H. B. McMahan, D. Ramage, K. Talwar, and L. Zhang.
>
> [25] See through gradients: Image batch recovery via gradinversion., H. Yin, A. Mallya, A. Vahdat, J. M. Alvarez, J. Kautz, and P. Molchanov.
>
> [Li’21] FedBN: Federated Learning on Non-IID Features via Local Batch Normalization, Xiaoxiao Li, Meirui Jiang, Xiaofei Zhang, Michael Kamp, Qi Dou.
>
> [Bonawitz’17] Practical secure aggregation for privacy-preserving machine learning, Bonawitz, Keith, et al.

---

### Decision · Program_Chairs · 2021-09-27

**Decision:**

Accept (Poster)

**Comment:**

The reviewers all agreed that this is an interesting paper and a worthy improvement over existing gradient inversion attacks against federated learning. I agree with the reviewers: the paper is clearly written and the many experimental results are worthy of publication. The paper will, at the very least, serve as an important motivation for further research on privacy in federated learning. I do caution the authors to take the suggestions made by the reviewers and the changes proposed in their rebuttal seriously and improve the manuscript in terms of related work and additional experiments.

It is also essential for the authors to implement what they promised in their rebuttal to the ethics reviewers in the final version of the paper. Critically, an experiment to demonstrate a potential fundamental defense mechanism and replacing human-face images in the paper's quantitative evaluation should be considered as "mandatory" revisions. I agree with the ethics reviewers that the authors' response helps assuage the ethical concerns (assuming that changes will be made to the final version).